# Exploration of the Use of Natural Compounds in Combination with Chemotherapy Drugs for Tumor Treatment

**DOI:** 10.3390/molecules28031022

**Published:** 2023-01-19

**Authors:** Jianping Wu, Yunheng Li, Qiaojun He, Xiaochun Yang

**Affiliations:** 1Center for Drug Safety Evaluation and Research, College of Pharmaceutical Sciences, Zhejiang University, Hangzhou 310058, China; 2Innovation Institute for Artificial Intelligence in Medicine of Zhejiang University, College of Pharmaceutical Sciences, Zhejiang University, Hangzhou 310058, China; 3Hangzhou Institute of Innovative Medicine, College of Pharmaceutical Sciences, Zhejiang University, Hangzhou 310058, China; 4Shandong (Linyi) Institute of Modern Agriculture, Zhejiang University, Linyi 276000, China

**Keywords:** chemotherapy, drug resistance, adverse reaction, natural compounds, synergistic interaction

## Abstract

Currently, chemotherapy is the main treatment for tumors, but there are still problems such as unsatisfactory chemotherapy results, susceptibility to drug resistance, and serious adverse effects. Natural compounds have numerous pharmacological activities which are important sources of drug discovery for tumor treatment. The combination of chemotherapeutic drugs and natural compounds is gradually becoming an important strategy and development direction for tumor treatment. In this paper, we described the role of natural compounds in combination with chemotherapeutic drugs in synergizing, reducing drug resistance, mitigating adverse effects and related mechanisms, and providing new insights for future oncology research.

## 1. Introduction

Tumors are currently one of the leading causes of human death worldwide, and the incidence and mortality rate of tumors is increasing worldwide, making tumors a huge risk factor for human health [1]. Treatment options for tumors include surgery, chemotherapy, radiation therapy, and immunotherapy. The use of chemical drugs to inhibit tumor progression remains the mainstay of treatment. However, as chemotherapy drugs continue to be used, tumors seem to become more and more resistant causing the killing effect of chemotherapy drugs to gradually lose efficacy [2]. In addition to this difficulty, the serious adverse effects of chemotherapy on patients are also a problem that chemotherapy drugs need to address urgently [3,4,5].

Natural compounds are an important source of drug discovery and a rich treasure trove of resources for human response to disease. Natural compounds such as paclitaxel, derivatives of camptothecin (docetaxel, irinotecan), and other antitumor drugs still play a pivotal role in the field of tumor treatment [6,7]. In addition, a variety of pharmacologically active natural compounds such as quercetin and curcumin also play a significant role in antitumor or combined therapeutic chemotherapy drugs against tumors, and in improving the quality of survival of patients. Therefore, natural compounds are an essential part of oncology drug research.

Here, we have compiled and summarized the literature on the beneficial effects of natural compounds in combination with chemotherapeutic drugs in cancer treatment in recent years to analyze the adjuvant role and feasibility of natural compounds as a future treatment for tumors, and to provide some research basis for selecting reasonable treatment strategies in oncology clinical trials to improve the overall survival rate and quality of life of patients.

## 2. Natural Compounds That Synergistically Enhance the Effects of Cancer Chemotherapy and Their Mechanisms

With the in-depth research on tumor therapeutic targets, chemotherapeutic drugs with different targets and pathways of action have started to adopt a combination therapy strategy to deal with the malignant development of tumors [8,9]. However, considering the more serious adverse effects of chemotherapeutic drugs, the combination of chemotherapeutic drugs with less toxic natural compounds that contribute to tumor management is gradually becoming one of the new strategies for tumor treatment. The specific synergistic mechanisms will be described from NF-κB, Hedgehog, LC3-I, LC3-II, Nrf2/HO-1, and other pathways (Table 1).

### 2.1. NF-κB

NF-κB, a specific transcription factor produced by B cells, binds to the enhancer sequence of the kappa light chain of activated B cells and plays an important role in promoting cell proliferation, inhibiting apoptosis, and cell migration [10]. Combined with previous studies related to NF-κB and cancer, NF-κB activation is one of the known important markers of cancer and is involved in the development and progression of many cancers, such as prostate cancer and bladder cancer [11,12]. Chronic inflammation and NF-κB can promote tumor growth and accelerate tumor malignant progression by activating reactive oxygen species that cause DNA damage and oncogenic mutations and promote inflammatory factors. Therefore, NF-κB may serve as a new avenue for natural compounds to synergize chemotherapeutic agents in the treatment of tumors [13].

Curcumin is a diketone natural compound extracted from the rhizome of turmeric and is widely used in the food industry as a natural pigment. Numerous studies have shown that curcumin has various physiological activities such as inhibiting tumor cell proliferation and promoting apoptosis [14]. Curcumin promotes tumor cell apoptosis by inhibiting NF-κB in breast cancer cell line MCF-7, pancreatic stellate cells, and liver cancer stem cells [15,16,17]. Curcumin may increase the activity of other chemotherapeutic agents such as paclitaxel. Paclitaxel is a classical chemotherapeutic drug that inhibits the G2/M phase of the cell cycle by stabilizing the homeostasis of intracellular microtubulin and reducing the depolymerization of dimeric microtubulin, thereby inhibiting the proliferation of cancer cells and promoting their apoptosis [18]. Curcumin increases paclitaxel activity by inhibiting NF-κB expression [19]. In addition, the combination of curcumin and doxorubicin significantly inhibits the proliferation and migration ability of AGS cells and promotes apoptosis in gastric cancer cells only at a concentration of 5 μg/mL [20]. A randomized clinical trial study showed that the combination of curcumin with melphalan and prednisone, therapeutic agents for multiple myeloma not suitable for transplantation, was effective in reducing the levels of NF-κB, VEGF, TNF-α, and IL-6 in patients, and the overall remission rate of patients was significantly increased [21]. Gambogic acid (GA) is a natural product extracted from the garcinia resin of the Garcinia hanburyi tree [22]. GA co-operates with cisplatin to increase the sensitivity of NSCLC cells to cisplatin by inhibiting NF-κB (p65 and p50), MAPK/ERK, MAPK/JNK, and promotes apoptosis in A549 and NCI-H460 cells [23].

### 2.2. Hedgehog

Hedgehog (HH) is a protein discovered in Drosophila with a key role in cell proliferation, differentiation, and survival [24]. However, aberrant activation of this pathway is associated with a variety of cancers, and two main hypotheses exist. One hypothesis suggests that the HH signaling pathway has an important role in the survival and proliferation of tumor cells themselves, and that evidence demonstrates that the HH signaling pathway is involved in Warburg-like glycolytic metabolism [25]. The other hypothesis speculates that the HH signaling pathway promotes the stromal cells surrounding the tumor through a paracrine form, which in turn affects the tumor cells [26]. Solamargine is a steroidal alkaloid from the traditional Chinese herb Solanum nigrum L., with anti-inflammatory and antitumor biological activities [27]. In the cisplatin-resistant lung cancer cell lines NCI-H1299 and NCI-H460, solamargine inhibited cell proliferation and promoted apoptosis by targeting SMO and thereby inhibiting the HH pathway. More importantly, the combination of solamargine and cisplatin showed a synergistic effect with each enhancing the other’s efficacy [28]. Sulforaphane, an isothiocyanate found in cruciferous vegetables, has been used in a variety of cancers for blocking the cellular G2/M phase transition leading to cell cycle arrest and apoptosis [29]. Combination treatment with sulforaphane and gefitinib dose-dependently inhibits the expression of SHH, SMO, and GLI1 and suppresses the proliferation of gefitinib-resistant lung cancer cells through the SHH signaling pathway [30].

### 2.3. LC3-I, LC3-II

Induction of ROS production by mitochondria of tumor cells leading to cell damage has been an important mechanism of chemotherapeutic drugs. However, autophagy of tumor cells can inhibit ROS production and weaken the killing effect of chemotherapeutic drugs, which is one of the reasons for the unsatisfactory effect of chemotherapeutic drugs [31,32]. Therefore, inhibition of autophagic flux in tumor cells leading to the accumulation of damaged mitochondria and ROS is one of the research directions to promote apoptosis in tumor cells [33]. Hederagenin is a pentacyclic triterpenoid found in a variety of medicinal plants and has a wide range of pharmacological effects including antitumor, anti-inflammatory, and antidepressant [34]. Wang Kun et al., found that hederagenin inhibited autophagy by increasing the conversion of LC3-I to LC3-II in lung cancer cells and that the combination of hederagenin with paclitaxel and cisplatin, respectively, could enhance their anticancer effects and play a synergistic role [35].

### 2.4. Nrf2/HO-1

Nrf2/HO-1 is an important signaling pathway that regulates redox and maintains intracellular homeostasis in mammalian cells. Nrf2-related pathway, closely related to iron death, has been shown to be important in promoting apoptosis in tumor cells. Tagitinin C is one of the active substances isolated from Tithonia diversifolia and has a wide range of anti-inflammatory, antitumor, and other pharmacological activities [36,37]. The combination of tagitinin C and erastin promotes apoptosis in HCT116 cells by further activating endoplasmic reticulum stress and enhancing iron death. Erastin works by inhibiting cystine-glutamate reversal causing iron death, while tagitinin C promotes cell death by upregulating HO-1 and promotes iron accumulation and ROS production mechanisms of action. However, the feasibility of this pathway is controversial, and some studies have shown that the addition of inhibitors of Nrf2-associated pathway proteins during chemotherapeutic drug treatment can instead effectively promote the therapeutic effects of chemotherapeutic drugs [38,39], with results inconsistent with the effects of tagitinin C. Ginkgetin is a flavonoid derived from Ginkgo biloba, and its incorporation with cisplatin can exert enhanced antitumor effects of cisplatin by promoting iron death, increasing ROS production, and inhibiting Nrf2/HO-1 [40]. Since the Nrf2/HO-1 pathway is involved in several processes such as oxidative stress and cellular detoxification, the mechanisms of which have not been clearly studied, and the mechanism of action of HO-1 protein has been controversial, Nrf2/HO-1 as the target of antitumor drugs needs to be studied more thoroughly, and related compounds need more specific and accurate mechanisms of action studies.

### 2.5. TMEM16A

TMEM16A is a calcium-activated chloride channel that is essential for maintaining cellular ion homeostasis and is highly expressed in various cancers such as prostate, lung, and colorectal cancers. It has been shown that TMEM16A inhibition can effectively reduce tumor growth, promote the sensitivity of tumor cells to chemotherapeutic agents, and improve overall patient survival [41]. Narirutin, a flavonoid isolated from Citrus unshiu, increased the antitumor effect of cisplatin in combination with cisplatin for lung cancer by dose-dependent inhibition of TMEM16A [42]. Homoharringtonine, an alkaloid isolated from the Cephalotaxaceae family, has been clinically shown to have antitumor effects [43]. Homoharringtonine inhibited TMEM16A in a dose-dependent manner and significantly inhibited the development of lung cancer at ex vivo levels [44]. In addition, theaflavin (tea polyphenol in black tea) and matrine (alkaloid in matrine) have also been shown to exert antitumor effects through TMEM16A [45,46]. TMEM16A is another potential antitumor target discovered in recent years, with the advantages of high safety and low toxicity, and its inhibitor combined with chemotherapy drugs may become a new therapeutic strategy for the treatment of TMEM16A high expression tumors in the future.

## 3. Natural Compounds That Reduce Tumor Drug Resistance and their Mechanisms

The development of resistance to chemotherapy in tumors is an inevitable and important issue, and statistics show that more than 90% of mortality in cancer patients is attributed to drug resistance, and the mechanisms by which it occurs are complex [47]. The mechanism of multi-drug resistance during chemotherapy can be attributed to the following reasons: (1) P-glycoprotein in cancer cells can excrete chemotherapeutic drugs from the cell, resulting in lower intracellular chemotherapeutic drug concentrations and reduced accumulation of chemotherapeutic drugs [48]. (2) Cancer cells enhance their DNA repair function mainly through nucleotide excision repair and the mismatch repair pathway to reduce apoptosis caused by DNA damage, thus increasing their resistance to platinum-based chemotherapy drugs [49,50]. (3) Mutation generation of key genes such as TP53 and drug target genes in cancer cells is also one of the important reasons why chemotherapeutic drugs lose their ability to kill [51,52]. We shed light on the mechanism of action of natural compounds in reducing tumor drug resistance (Table 2).

### 3.1. PAFR

PAFR, the acting receptor for PAF is a G protein-coupled receptor that is closely associated with platelet aggregation, inflammation, and nerve damage [53]. In recent years, PAFR has been found to be equally associated with the progression of a variety of tumors, and some studies have shown that PAFR is upregulated and promotes malignant progression in non-small cell lung cancer, esophageal squamous carcinoma, ovarian cancer, and other tumors [54,55]. In non-small cell lung cancer, PAFR initiates a positive feedback loop between PAFR and STAT3 to promote tumor growth and metastasis [56]. PAFR-regulated PI3K/AKT pathway activation stimulates tumor progression in esophageal squamous carcinoma [57]. In this regard, Aponte et al., found that the PAF/PAFR pathway promotes the proliferation and invasion of ovarian cancer through tyrosine phospho-EGFR/Src/FAK/paxillin [55].

Ginkgolide B is a natural compound derived from the traditional Chinese medicine Ginkgo, which has a strong antagonistic ability against platelet-activating factor and is the strongest compound in nature that specifically antagonizes PAFR [58]. It has been shown that in the concentration range where ginkgolide B does not produce cytotoxicity, gemcitabine in combination with ginkgolide B can enhance the effect of gemcitabine in killing resistant pancreatic cancer cells by inhibiting the PAFR/NF-κB pathway and reduce the resistance of pancreatic cancer to gemcitabine [59]. The important role of PAFR in oral cancer was also confirmed by the study of Kohei Kawasaki et al. Cisplatin in combination with ginkgolide B inhibited PAFR and the phosphorylation levels of its downstream signaling pathways ERK and Akt, and promoted the expression of cleaved caspase-3, leading to apoptosis and increasing the sensitivity of oral cancer cells to cisplatin treatment [60]. The treatment strategy of cisplatin in combination with ginkgolide B has the same effect of reducing tumor growth and increasing drug sensitivity in ovarian cancer [61]. Ichim, G et al. suggested that apoptosis induced after chemotherapy or radiotherapy is twofold, promoting apoptosis to induce tumor cell death while also inducing further tumorigenesis [62]. Furthermore, the latest research results found that PAF is produced during chemotherapy and radiotherapy for cancer treatment, and PAF has an oncogenic function when combined with PAFR. Therefore, natural inhibitors of PAFR may become one of the new directions of tumor treatment in the future [63,64].

### 3.2. Pin1

Prolyl isomerase 1 (Pin1) is a peptidyl-prolyl cis/trans isomerase that regulates the biological functions of a variety of proteins through conformational changes and has a key role in Alzheimer’s disease and several cancers [65]. Functionally, in addition to Pin1 activating various cancer pathways such as Raf/MEK/ERK, PI3K/Akt, Wnt/β-catenin, NF-κB. Pin1 drives pro-connective tissue proliferation and immunosuppressive TME and promotes tumor malignancy and drug resistance by acting on stromal cells such as CAF and by acting on pS929-HIP1R to induce endocytosis and degradation of PD-L1 and ENT1 in cancer cells. [66]. Kazuhiro Koikawa et al. found that Pin1 was highly expressed in pancreatic ductal adenocarcinoma and cancer-associated fibroblasts (CAF), and that Pin1 inhibitor synergized with PD1 inhibitor αPD1 to promote apoptosis and significantly reduce tumor growth in human and KPC PDAC-like organoids in GDA mice [67]. Therefore, targeting Pin1 offers a unique and promising approach to eradicate this deadly cancer.

Juglone is a natural naphthoquinone found in the walnut tree. Juglone and its derivatives are inhibitors of Pin1 and are effective in reducing chemotherapy resistance due to cancer treatment. [68]. Sajadimajd S et al. found that, in trastuzumab SKBR3 cells, juglone could induce cell apoptosis, inhibit cell proliferation, colony formation, and migration, and promote the reduction of drug resistance by inhibiting Pin1 and Notch1 [69]. Similarly, Yun H et al. also found that juglone significantly enhanced trastuzumab-induced FAS downregulation and cell death in metastatic breast cancer BT474 cells. In addition, trastuzumab in combination with gene silencing or juglone increased cleaved poly(ADP-ribose) polymerase and DNA fragmentation, thereby increasing the sensitivity of trastuzumab [70]. For estrogen receptor alpha-positive breast cancer, juglone dose-dependently inhibits TPA-induced tumor cell transformation by reversing the TPA-induced rise in E2F-4 and Egr-1 and downregulating LC-3, thereby enhancing the sensitivity of tamoxifen-resistant cells MCF-7 to tamoxifen [66]. In addition to juglone, epigallocatechin-3-gallate (EGCG), all-trans retinoic acid (ARTA), and arsenic trioxide (ATO) have shown good efficacy as inhibitors of Pin1 in reducing tumor resistance.

### 3.3. P-Glycoprotein

P-glycoprotein, also known as multi-drug resistance protein 1 (MDR1), is a superfamily of ATP binding box (ABC) transporter proteins and an ATP-dependent drug efflux pump, which can reduce the accumulation of intracellular drugs and mediate the generation of cell drug resistance [71]. In addition, multidrug resistance-associated protein 1, multidrug resistance-associated protein 2, and breast cancer resistance protein also generate drug resistance by increasing the efflux of chemotherapeutic drugs generally considered to be the main cause of MDR [72]. Schisandrin B was isolated from Schisandra chinensis, a traditional Chinese medicine, and has antioxidant and antitumor activities [73]. Schisandrin B reduces tumor drug resistance by decreasing p-glycoprotein expression in a variety of tumors [74]. Schisandrin B inhibits the expression and activity of p-glycoprotein in doxorubicin-resistant breast and ovarian cancer cells, thereby enhancing the intracellular accumulation of doxorubicin and reducing the generation of drug resistance [75]. In addition, a study showed that Schisandrin B reverses the resistance of K562/ADR, KBv200, and MCF-7/Adr to paclitaxel, anthracycline, and vincristine by direct physical interaction with p-glycoprotein [76]. Caffeic acid is a kind of phenolic acid widely found in plants. TENG Y-N et al., found that caffeic acid significantly reversed the resistance of tumor cells to vincristine, paclitaxel, and doxorubicin, and increased the percentage of apoptosis in tumor cells [77]. In addition, the prenylated flavonoid from Tephrosia purpurea, glabratephrin, has been shown in recent years to enhance the efficacy of doxorubicin by reducing the affinity of doxorubicin for p-glycoprotein and preventing its efflux without affecting p-glycoprotein expression in triple-negative breast cancer cells [78]. By reversing p-glycoprotein-mediated resistance, it can be used as a research direction for chemotherapeutic drug sensitizers and provide a safe and effective strategy for treating tumor cells that develop resistance to drugs.

### 3.4. PI3K/Akt

Phosphatidylinositol-3 kinase (PI3K)/Akt pathway is one of the important intracellular signal transduction pathways, which plays a key role in glucose uptake and metabolism [79]. This pathway is highly activated in tumors, which produces favorable conditions for the growth and proliferation of tumor cells, and is one of the important reasons for the development of drug resistance in tumors [80]. Quercetin is a polyphenolic flavonoid compound widely distributed in fruits and vegetables, with a variety of pharmacological activities such as anti-inflammatory and antioxidant [81]. In vivo and in vitro models of docetaxel resistance, quercetin combined with docetaxel improved the inhibition of cell proliferation, metastasis, and invasion, and reversed docetaxel resistance through the PI3K/AKT pathway [82]. Isorhamnetin, which is also a flavonoid with quercetin, has also been shown to have similar effects [83]. Toosendanin is a triterpenoid compound extracted from Melia toosendan Sieb. et Zucc with ascaris removal and antibacterial activities [84]. Toosendanin combined with doxorubicin significantly promoted the apoptosis of drug-resistant MCF-7 cells and inhibited the phosphorylation of AKT at the non-cytotoxic concentration of toosendanin. Toosendanin and doxorubicin alone had a weak inhibitory effect on tumor growth, while the combined administration of toosendanin and doxorubicin had a 90% inhibitory effect on tumor volume [85]. Matrine, the main active substance extracted from Matrine, can also decrease the drug resistance of MCF-7 cells by up-regulating PI3K/AKT and down-regulating the phosphorylation level of AKT through the negative regulatory factor PTEN [86]. Apigenin is a natural flavonoid with broad-spectrum biological properties including antioxidant, anti-inflammatory, anti-cancer, and neuroprotective effects [87]. In a study of gemcitabine-resistant pancreatic cancer cells, apigenin combined with gemcitabine was found to block the cell cycle of drug-resistant cells, downregulate gemcitabine-induced p-Akt, and induce apoptosis in tumor cells [88].

### 3.5. Notch

In recent years, epithelial–mesenchymal transition (EMT) has been shown to be an important factor in the development of drug resistance in tumor cells, and the mechanisms that help tumor cells to develop drug resistance are mainly attributed to overexpression of drug transport proteins (p-glycoprotein, multidrug resistance associated protein 1, etc.) and inhibition of tumor cell apoptosis [89]. Notch is one of the important pathways in the epithelial–mesenchymal transition and is involved in the development of drug resistance in tumor cells. In Notch-overexpressing breast cancer cells, positive regulation of SLUG by Notch ^IC^ activation leads to suppression of E-cadherin, thus allowing EMT in breast cancer cells [90]. Furthermore, the Notch pathway has been shown to act against drug resistance in various cancers such as prostate cancer and lung cancer [91,92,93]. Notch inhibitors represented by curcumin play an adjuvant antitumor role in cancer stem cells [94,95,96,97].

### 3.6. TGF-β

TGF-β is also a key pathway in the epithelial–mesenchymal transition and is involved in the metastasis and invasion of tumor cells [98]. MHP-1, a newly isolated polysaccharide from Mortierella hepialid, attenuated topiramate resistance in breast cancer cells and inhibited the process of EMT by inhibiting the TGF-β pathway [99].

### 3.7. MGMT

Causing tumor cell death by inducing DNA damage is the main mechanism by which some chemotherapeutic drugs work, while tumor cells rescue themselves from the damage by their own DNA repair function thus creating resistance to chemotherapeutic drugs. O (6)-methylguanine-DNA methyltransferase (MGMT) is an important transferase in the DNA repair process for the elimination of toxic and premutagenic DNA adduct O^6^-methylguanine from cells [100]. Lipoic acid, disulfide-containing substance, is a naturally occurring cofactor for the mitochondrial enzymes pyruvate dehydrogenase and alpha-ketoglutarate dehydrogenase [101]. It was reported that lipoic acid not only increased alkylating agent N-methyl-N-nitrosourea-induced O^6^-MeG lesions by inhibiting MGMT, but also attenuated temozolomide resistance in colorectal cancer cells HCT116 [102].

### 3.8. EGFR

Epidermal growth factor receptor (EGFR) belongs to a family of cell surface receptor tyrosine kinases whose wild-type signaling contributes to the proliferation of tumor cells, evades apoptosis, and promotes tumor proliferation and invasion. EGFR-tyrosine kinase inhibitors (TKI) such as gefitinib target EGFR for antitumor effects, but EGFR mutations (Such as T790M or S492R mutations) are an important cause of tumor cell generation [103]. A study reported that gambogic acid in combination with EGFR-TKI effectively inhibited EGRF-T790M mutated lung adenocarcinoma cell lines and suppressed tumor volume growth in vivo [104]. In addition, Formononetin, as an inhibitor of EGFR, inhibited EGFR-Akt signaling by binding to the ATP-binding pocket region of both wild-type and mutant EGFR, promoting the ubiquitination and degradation of Mcl-1, thereby inhibiting the proliferation of non-small cell lung cancer [105].

## 4. Natural Compounds That Attenuate Adverse Effects of Chemotherapy and their Mechanisms

Serious adverse reactions caused by chemotherapeutic drugs are one of the main reasons affecting the treatment outcome of tumor patients. Death of some tumor patients is associated with the occurrence of adverse reactions. We have elaborated on the natural compounds in combination with chemotherapeutic drugs that can reduce the adverse effects and help in the treatment of tumors (Table 3).

### 4.1. Neurotoxicity

Oxaliplatin inhibits tumor cell proliferation by forming DNA-platinum adducts with DNA [106]. Some chemotherapeutic drugs can cause peripheral neuropathy to varying degrees, the most serious of which are paclitaxel and oxaliplatin [107]. Mitochondrial dysfunction is considered to be one of the important mechanisms of peripheral neuropathy [108]. It has been found that tanshinone IIA, an active substance extracted from the famous Chinese medicine Salvia miltiorrhiza, can inhibit oxaliplatin-induced ROS increase in mouse neuroma cell line N2a, thus achieving mitochondrial protection. In addition, tanshinone IIA could alleviate oxaliplatin-induced peripheral neuropathy by promoting autophagy through PI3K/Akt/mTOR signaling pathway. Tanshinone IIA in the non-cytotoxic concentration range can inhibit the pro-apoptotic effect of oxaliplatin on mouse neuroma cell line N2a and reduce oxaliplatin-induced neurotoxicity in rats [109]. Natural antioxidants thymoquinone and geraniol can reduce cisplatin-induced neurotoxicity by inhibiting the expression of apoptosis-related proteins (p53, MAPK, etc.) without affecting the killing effect of cisplatin on breast cancer MCF-7 cells [110]. Berberine, an isoquinoline alkaloid, has been shown to have neuroprotective effects on doxorubicin-induced neuroinflammation by increasing brain AchE activity and reducing neuronal apoptosis caused by oxidative stress [111].

### 4.2. Myelosuppression

Myelosuppression is one of the main adverse reactions of many chemotherapy drugs, including cyclophosphamide, paclitaxel, pemetrexed, and gemcitabine, which seriously affects the therapeutic effect of chemotherapy drugs [112,113]. Chemotherapy-induced bone marrow suppression can be manifested as neutropenia, leukopenia, and anemia [114]. Ginsenoside Rg3, a tetracyclic triterpene saponin extracted from Red ginseng, has been shown to selectively inhibit tumor cell invasion and metastasis [115]. However, a study on the QT prolongation induced by the anti-thyroid cancer drug vandetanib showed that ginsenoside Rg3 combined with vandetanib could increase the inactivating current of hERG Kchannel, thereby reversing the QT prolongation [116].

### 4.3. Gastrointestinal Toxicity

Diarrhea is one of the main adverse reactions of chemotherapy drugs such as 5-fluorouracil, irinotecan, and celecoxib [117,118,119]. Mild chemotherapy-induced diarrhea can interfere with the process and effect of cancer treatment. Severe diarrhea can cause dehydration, electrolyte imbalance, and nutritional deficiency, which are associated with premature death in 5% of cancer patients [120]. Hesperidin is a natural flavonoid widely found in fruits and flowers, with a variety of pharmacological activities such as anti-inflammatory, antioxidant, cardiovascular protection, and antitumor [121]. Oral administration of hesperidin 20 mg/kg and 100 mg/kg significantly reduced irinotecan-induced diarrhea in CT-26 tumor-bearing immune mice and reduced the risk of severe diarrhea. In addition, hesperidin inhibited the expression of inflammatory factors in intestinal tissues and, more importantly, hesperidin in combination with irinotecan could exert more significant antitumor effects by negatively regulating STAT3 [122].

### 4.4. Cardiotoxicity

Anthracycline antineoplastic agents are an important part of chemotherapeutic drugs and are widely used in hematologic malignancies and solid tumors, but the typical use of anthracycline antineoplastic agents is associated with cardiotoxicity, which seriously affects the progress of subsequent treatment [123]. Calycosin is an active ingredient in Astragalus membranaceus, which has various pharmacological activities such as anti-inflammatory, antioxidant, anti-cancer, and cardiovascular protection. [124]. Calycosin at a concentration of 20 μg/mL significantly inhibited doxorubicin-induced LDH, ROS, and mitochondrial damage in H9c2 cells, while in vitro and in vivo experiments showed that calycosin attenuated the cardiotoxicity of doxorubicin by inhibiting NLRP3-cystatin-1-GSDMD pathway-mediated cardiomyocyte scorching [125]. Zhai J. et al. found that calycosin attenuated doxorubicin-induced apoptosis and ROS production in H9c2 cells via the Sirtuin-NLRP3 pathway [126]. Calycosin reduces doxorubicin-induced pericardial edema and morphological changes while increasing embryo viability in a zebrafish model [127]. Colchicine, an alkaloid originally extracted from the lily family Colchicum, is found in corn, seeds, and flowers [128]. 5-Fluorouracil has some cardiotoxicity, resulting in cardiac electrophysiological abnormalities, including ST-segment elevation and significant prolongation of QRS duration [129]. The combination of the two can reduce cardiac abnormalities and damage by reducing oxidative stress in cardiomyocytes, increasing the total antioxidant capacity of the heart, and reducing cardiotoxicity caused by 5-fluorouracil treatment [130]. In addition to these compounds, several studies have found that quercetin, silymarin, asiatic acid, tanshinone IIA, and many other compounds can have some protective effects in the presence of cardiotoxicity from chemotherapy [131,132,133,134,135]. The curcumin mentioned in the previous part of the article also inhibited doxorubicin-induced cardiomyocyte scorching, but this result is controversial [136].

### 4.5. Nephrotoxicity

Curcumin, thymoquinone, and As_2_O_3_ all attenuated cisplatin-induced renal fibrosis and reduced tubular injury, renal α-SMA, and renal fibrosis scores [137,138]. However, clinical studies have shown that the use of As_2_O_3_ in the treatment of relapsed or refractory acute promyelocytic leukemia and multiple myeloma can cause renal damage, such as increased levels of serum creatinine, blood urea nitrogen, and protein urea concentration [139]. Therefore, whether As_2_O_3_ can be used as a regimen to reduce chemotherapy nephrotoxicity remains to be investigated. Resveratrol, a natural antioxidant widely used in cardiovascular disease and anti-aging, also has a protective effect in reducing chemotherapy-induced nephrotoxicity. In an experiment in mice treated with cisplatin for one week, resveratrol reduced the activation of the cisplatin-associated p53 acetylation and apoptosis pathways by increasing Sirt1, thereby increasing the glomerular filtration rate in mice [140].

### 4.6. Hepatotoxicity

5-FU, oxaliplatin, and irinotecan for tumors are thought to cause liver damage by increasing the production of reactive oxygen species in hepatocytes [141,142]. [10]-Gingerol is a stimulating compound extracted from the daily seasoned food ginger, with various pharmacological activities such as antioxidant and antitumor [143,144]. Although [10]-Gingerol in combination with doxorubicin did not show a significant difference in tumor volume at day 28 in tumor-bearing mice compared to doxorubicin alone, the combination treatment reduced chemotherapy-induced weight loss and hepatotoxicity [145].

## 5. Conclusions 

In summary, the therapeutic strategy of combining natural compounds with chemotherapeutic drugs can effectively enhance the tumor-killing effect of chemotherapeutic drugs, reduce the development of drug resistance in tumor cells, and alleviate the serious side effects of chemotherapeutic drugs on patients, which has a positive effect on tumor treatment. Natural compounds, especially those used in traditional Chinese medicine, have been used in human diseases for thousands of years, providing many active products for human beings, and also providing certain reference for the subsequent clinical research of combined chemotherapy drugs. A natural compound often has multiple pharmacological activities, for example, tanshinone IIA can not only attenuate oxaliplatin-induced neurotoxicity, but also inhibit doxorubicin-induced cardiotoxicity, hepatotoxicity, and nephrotoxicity. Natural compounds in combination with chemotherapeutic agents are often more effective in their antitumor effects for reasons that are often not singular, and their beneficial results may be due to a combination of multiple mechanisms. Such multiple effects further confirm the feasibility of combining natural compounds with chemotherapeutic agents in the treatment of cancer. We can see that natural compounds have considerable potential to deal with the adverse reactions caused by chemotherapy, which can effectively alleviate the toxic effects caused by chemotherapy, assist the follow-up treatment of patients, and improve the quality of life (Figure 1).

However, natural compounds may have some disadvantages, such as poor water solubility and unfavorable pharmacokinetics, and these deficiencies may affect the combined effect of natural compounds with chemotherapeutic drugs. Therefore, a series of methods such as structural modification of natural compounds or improvement of drug delivery can strongly break the dilemma of combining natural compounds with chemotherapeutic drugs, providing a clear direction for future research to realize the combination of the two. In addition, the discovery of potentially useful natural compounds is also an issue that needs to be addressed urgently. Although Chinese herbal medicine has a long history of use with remarkable effects, the active monomers on which they work have not been fully studied and many of these active compounds have not been discovered, therefore this area may provide some ideas for finding such natural compounds.

In summary, we personally believe that natural compounds for tumor combination therapy have great potential in synergizing chemotherapeutic agents, reducing tumor cell resistance, and alleviating adverse effects. The therapeutic strategy of natural compounds combined with chemotherapeutic drugs will be gradually applied in clinical trials and become a new exploration to defeat tumors.

## Figures and Tables

**Figure 1 molecules-28-01022-f001:**
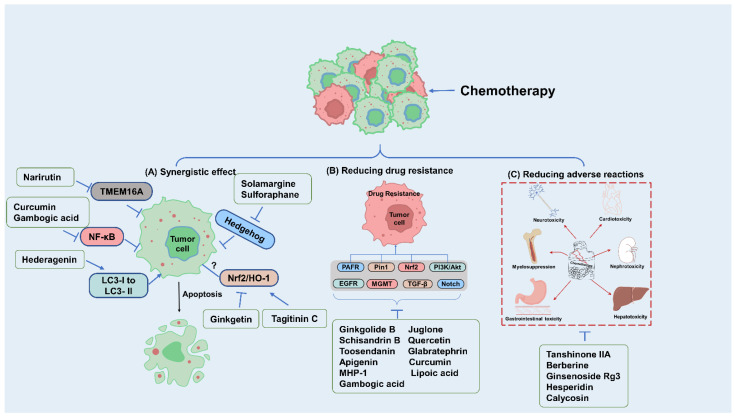
Natural compounds in combination with chemotherapeutic agents enhance therapeutic efficacy by synergizing, reducing tumor resistance, and mitigating adverse effects of chemotherapy. (**A**) Natural compounds exert synergistic effects in combination with chemotherapeutic treatment to promote apoptosis of tumor cells and enhance antitumor effects. (**B**) Natural compounds alleviate tumor resistance produced by chemotherapy. (**C**) Natural compounds significantly alleviate the toxic damage caused by chemotherapy to multiple organs.

**Table 1 molecules-28-01022-t001:** Natural compounds that synergistically enhance the effects of cancer chemotherapy and their mechanisms.

Natural Products	Molecular Structure	Cancer	Combined Chemotherapy Drugs	Mechanism
Curcumin	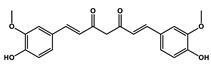	Gastric cancer	Paclitaxel, doxorubicin, melphalan, and prednisone	NF-κB
		Multiple myeloma		
Gambogic acid	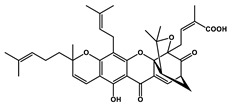	Non-small cell lung cancer	Cisplatin	NF-κB
Solamargine	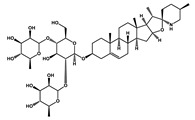	Non-small cell lung cancer	Cisplatin	Hedgehog
Sulforaphane	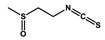	Non-small cell lung cancer	Gefitinib	Hedgehog
Hederagenin	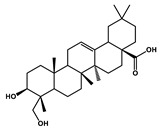	Lung cancer	Paclitaxel, cisplatin	LC3-I, LC3-II
Tagitinin C	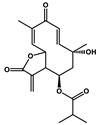	Colorectal carcinoma	Erastin	Nrf2/HO-1
Ginkgetin	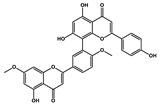	Non-small cell lung cancer	Cisplatin	Nrf2/HO-1
Narirutin	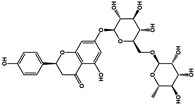	Lung adenocarcinoma	Cisplatin	TMEM16A

**Table 2 molecules-28-01022-t002:** Natural compounds that reduce tumor drug resistance and their mechanisms.

Natural Products	Molecular Structure	Cancer	Combined Chemotherapy Drugs	Mechanism
Ginkgolide B	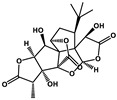	Pancreatic cancer	Gemcitabine, cisplatin	PAFR
Oral cancer
Juglone	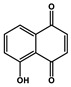	Metastatic breast cancer	Trastuzumab	Pin1, Notch
Estrogen receptor Alpha-positive breast cancer	Tamoxifen
Schisandrin B	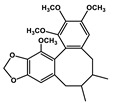	Breast cancer, Ovarian cancer	Doxorubicin, paclitaxel, anthracycline, and vincristine	P-glycoprotein
Caffeic acid	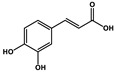	Cervical cancer	Vincristine, paclitaxel, and doxorubicin	P-glycoprotein
Glabratephrin	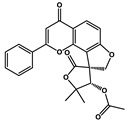	Triple-negative breast cancer cells	Doxorubicin	P-glycoprotein
Quercetin	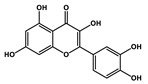	Prostate cancer	Docetaxel	PI3K/Akt
Isorhamnetin	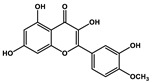	Prostate cancer	Docetaxel	PI3K/Akt
Toosendanin	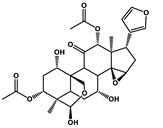	Breast cancer	Doxorubicin	PI3K/Akt
Matrine	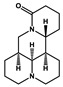	Breast cancer	Doxorubicin	PI3K/Akt
Apigenin	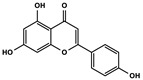	Pancreatic cancer	Gemcitabine	PI3K/Akt
Curcumin	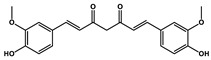	Colorectal cancer	Oxaliplatin	Notch
MHP-1	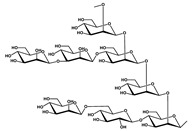	Breast cancer	Topotecan	TGF-β
Lipoic acid	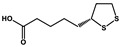	Colorectal cancer	Temozolomide	MGMT
Gambogic acid	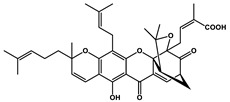	Non-small cell lung cancer	Gefitinib	EGFR
Formononetin	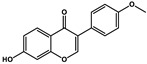	Non-small cell lung cancer	Osimertinib	EGFR

**Table 3 molecules-28-01022-t003:** Natural compounds that attenuate adverse effects of chemotherapy and their mechanisms.

Natural Products	Molecular Structure	Combined Chemotherapy Drugs	Mechanism
Tanshinone IIA	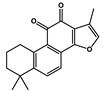	Oxaliplatin	Neurotoxicity reduction
Thymoquinone	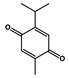	Cisplatin	Neurotoxicity reduction
Geraniol	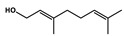	Cisplatin	Neurotoxicity reduction
Berberine	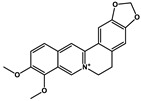	Doxorubicin	Neurotoxicity reduction
Ginsenoside Rg3	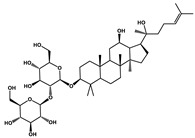	Vandetanib	Myelosuppression reduction
Hesperidin	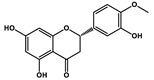	Irinotecan	Gastrointestinal toxicity reduction
Calycosin	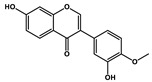	Doxorubicin	Cardiotoxicity reduction
Colchicine	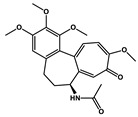	5-FU	Cardiotoxicity reduction
Quercetin	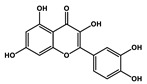	Cisplatin	Cardiotoxicity reduction
Silymarin	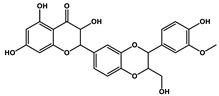	5-FU	Cardiotoxicity reduction
Asiatic Acid	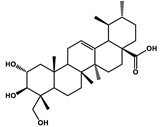	Doxorubicin	Cardiotoxicity reduction
Resveratrol	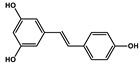	Cisplatin	Nephrotoxicity reduction
[10]-Gingerol	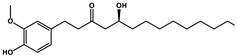	Doxorubicin	Hepatotoxicity reduction

## Data Availability

Not applicable.

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
