# Peer review of "Exploration of the Use of Natural Compounds in Combination with Chemotherapy Drugs for Tumor Treatment"

_molecules, 2023, doi:10.3390/molecules28031022_

Round 1

Reviewer 1 Report

Wu et al. conducted a review titled “A new strategy for tumor therapy: the application of natural compounds combined with chemotherapy drugs in tumor treatment”. This review described the beneficial effects of natural compounds in combination with chemotherapeutic drugs in cancer treatment. 

1.     The authors mentioned that this review summarized the effects of natural compounds  in combination with chemotherapeutic drugs in cancer treatment, and the authors summarize NF-κB and Hedgehog pathways as synergistic mechanisms. But current main studies believe natural compounds acting as chemotherapeutic drug sensitizers. And the authors should summarize all combination of natural compound and chemotherapy drugs. 

2.     The authors summarize PAFR, Pin1, P-gp and PI3K/Akt pathways as drug resistance mechanisms. However, prevailing mechanisms of chemoresistance are classified into seven phases: drug flux, DNA damage repair, cell death inhibition, epithelialmesenchymal transition (EMT), drug target alteration, drug inactivation, and epigenetics, and notably, drug flux is the most concerned issue in this topic. Also shoukd summarize what kind of drugs are affected by these mechanism.

Author Response

Comment 1: The current main studies believe natural compounds acting as chemotherapeutic drug sensitizers. And the authors should summarize all combination of natural compounds and chemotherapy drugs.

Response: We thanked the reviewer for raising this question. In recent years, a large number of articles have indeed been written on the use of natural compounds as sensitizers for chemotherapeutic drugs. This is a very interesting research direction. However, natural compounds are multi-targeted and multi-effective in nature, and they often exert specific pharmacological activities through different molecular mechanisms, which means that they may have multiple roles in combination with chemotherapeutic agents for the treatment of tumors. We have cited numerous articles in chapters 2 and 4 of our article on the synergistic treatment of tumors and mitigation of toxic reactions induced by chemotherapeutic agents with natural compounds, which also suggests that natural compounds have multiple identities in combination chemotherapy. We have supplemented the table with chemotherapeutic agents related to the action of natural compounds. Due to a large number of compounds in this category, we elaborate on typical compounds among them.

Comment 2: The authors summarize PAFR, Pin1, P-gp and PI3K/Akt pathways as drug resistance mechanisms. However, prevailing mechanisms of chemoresistance are classified into seven phases: drug flux, DNA damage repair, cell death inhibition, epithelial‐mesenchymal transition (EMT), drug target alteration, drug inactivation, and epigenetics, and notably, drug flux is the most concerned issue in this topic. Also should summarize what kind of drugs are affected by these mechanism.

Response: We are grateful for your kind question. As you pointed out, we also address the mechanisms of tumor resistance in the first paragraph of chapter2, where our perspective is detailed in terms of the role of natural compounds on the classical mechanisms involved as well as on the newly discovered resistance-related targets in recent years. For the other aspects, they are not the main focus of this article as they are not the focus of the exposition. As above, we have supplemented the table with chemotherapeutic agents related to the action of natural compounds.

Reviewer 2 Report

The literature review on the basis of which the authors wrote this review consists of 113 references published from 2005 to 2022, of which about 50% were published in the last 3 years indicating that the synergistic effects of combination therapy using herbal medicines and anticancer drugs have attracted much attention across the medical community. By searching only a few databases (PubMed, Embase, WoS, and Cochrane Library) it is possible to find number of articles and review papers dealing with this issue.

Considering the large number of published studies that are related and that review and summarize new facts related to this issue, unfortunately I do not see what new and specific information are given in this manuscript regarding this issue.

The title should definitely be changed, because authors cannot talk about a new strategy. It is also necessary to revise chapter 5. (Summary) and explain in more detail why exactly curcumin was singled out of all the natural compounds described.

Author Response

Reviewer 2
Comment 1: Considering the large number of published studies that are related and that review and summarize new facts related to this issue, unfortunately I do not see what new and specific information are given in this manuscript regarding this issue.

Response: Thank you for your valuable comments on our article. The increasing number of articles related to the use of natural compounds for the treatment of tumors in recent years
demonstrates the great potential of natural compounds in the fight against tumors. By searching PubMed and Web Of Science, we have found more articles describing the influence of natural compounds on the mechanism of action against tumor drug resistance. However, the thematic content of our articles did not agree with them. 1. There are not many review articles on natural compounds synergistic with chemotherapeutic drugs for the treatment of tumors, and most of the review articles are devoted to the elaboration of natural compounds as sensitizers. Here, we consider the natural compounds that can significantly enhance the antitumor effect in combination with chemotherapeutic agents at concentrations of compounds without cytotoxic (antitumor effect) as the type of sensitizer. In contrast, the natural compounds that can be combined with chemotherapeutic agents at concentrations of cytotoxic drugs as the type of synergistic effect, which is one of our special points. 2. We addressed the multiple identities of natural compounds in combination therapy. The role and prospects of natural compounds as synergists, sensitizers, and mitigators of adverse effects are comprehensively described. Moreover, there are few reviews related to the use of natural compounds to mitigate adverse effects of chemotherapy, which I think is another major highlight of our article.

Comment 2: The title should definitely be changed, because authors cannot talk about a new strategy. It is also necessary to revise chapter 5. (Summary) and explain in more detail why exactly curcumin was singled out of all the natural compounds described.

Response: Thank you for pointing out that we have a problem. The strategy of combining natural compounds with chemotherapeutic agents has been studied extensively, so it is indeed inappropriate to use the title of this article as a new strategy, and we appreciate your reminder. Our thought was that there was not much exploration of the strategy of combining natural compounds with chemotherapeutic agents in clinical trials, and we wanted to use the title of new strategy as a new direction in clinical trials. But the whole thing was not carefully considered. We have revised the title.

The description of curcumin is the result of the author's careless thinking. The purpose of this section is to illustrate the great potential of natural compounds and the subsequent research directions by showing the safety and multi-effectiveness of curcumin and the necessity of structural modification of natural compounds for actual clinical use. Revisions have been made.

Reviewer 3 Report

In this review article, the authors presented the strategy to improve chemotherapy by combining chemotherapeutic drugs and natural compounds. The role of natural compounds in such combination have been discussed in details regarding its synergistic effect, reduction of drug resistance and elimination of adverse effects. The article gives a comprehensive historical and scientific perspective on the field that has recently boomed. Overall, I find this review article well-structured and the topic covered is complete. From my perspective, it is suitable for publication in its present form.

Author Response

Reviewer 3
Comment 1: In this review article, the authors presented the strategy to improve chemotherapy by combining chemotherapeutic drugs and natural compounds. The role of natural compounds in such combination have been discussed in details regarding its synergistic effect, reduction of drug resistance and elimination of adverse effects. The article gives a comprehensive historical and scientific perspective on the field that has recently boomed. Overall, I find this review article well-structured and the topic covered is complete. From my perspective, it is suitable for publication in its present form.

Response: Thank you for your comments and suggestions. We will follow up with detailed changes to the language and content details, and thank you again for your comments.

Round 2

Reviewer 1 Report

The authors have performed additional modification, but the current structure and organization are still not enough to support the conclusion. Unfortunately, I can't find new information in this manuscript.  

Author Response

Thank you for your comprehensive and insightful suggestions, and we agree with you very much. We are sorry that we were not able to make a comprehensive revision based on your suggestions because we were unfortunately caught COVID-19 during the last revision. This time, we have provided more detailed information on the mechanisms of tumor drug resistance and the related literature, and have added more information on the study of natural compounds in combination with chemotherapeutic agents, such as tumor epithelial mesenchymal transition, DNA damage repair and related gene mutations, as well as more cutting-edge research targets in terms of synergistic effects. Thank you again for your valuable comments and we  wish you good health and success in 2023.

Reviewer 2 Report

Dear authors, I suggest that you modify the title by shortening it: “Exploration of the use of natural compounds in combination with chemotherapy drugs for tumour treatment" With regards,  

Author Response

Thank you for your valuable comments. We have revised the title according to your suggestion. Thank you again for your guidance. We wish you good health and all the best in your career in 2023.